# Reduction in Rubella Virus Active Cases among Children and Adolescents after Rubella Vaccine Implementation in Tanzania: A Call for Sustained High Vaccination Coverage

**DOI:** 10.3390/vaccines10081188

**Published:** 2022-07-27

**Authors:** Fausta Michael, Mariam M. Mirambo, Dafrossa Lyimo, Furaha Kyesi, Delfina R. Msanga, Georgina Joachim, Honest Nyaki, Richard Magodi, Delphius Mujuni, Florian Tinuga, Ngwegwe Bulula, Bonaventura Nestory, Dhamira Mongi, Ahmed Makuwani, Betina Katembo, William Mwengee, Alex Mphuru, Nassor Mohamed, David Kayabu, Helmut Nyawale, Eveline T. Konje, Stephen E. Mshana

**Affiliations:** 1Ministry of Health, Immunization and Vaccine Development Program, Dodoma P.O. Box 743, Tanzania; fausta.selemani@afya.go.tz (F.M.); dafrossa.lyimo@afya.go.tz (D.L.); furaha.kyesi@afya.go.tz (F.K.); georgina.temba@afya.go.tz (G.J.); honest.nyaki@afya.go.tz (H.N.); richard.magodi@afya.go.tz (R.M.); delphinus.rugemalila@afya.go.tz (D.M.); florian.tinuga@afya.go.tz (F.T.); ngwegwe.bulula@afya.go.tz (N.B.); bonaventura.muhindi@afya.go.tz (B.N.); dhamira.mongi@afya.go.tz (D.M.); ahmed.makuwani@afya.go.tz (A.M.); 2Department of Microbiology and Immunology, Weill Bugando School of Medicine, Catholic University of Health and Allied Sciences, Mwanza P.O. Box 1464, Tanzania; helmut.nyawale@bugando.ac.tz (H.N.); mshana72@bugando.ac.tz (S.E.M.); 3Department of Pediatrics and Child Health, Weill Bugando School of Medicine, Catholic University of Health and Allied Sciences, Mwanza P.O. Box 1464, Tanzania; deromsah@gmail.com; 4National Public Health Laboratory, Dar es Salaam P.O. Box 9083, Tanzania; betina.katembo@afya.go.tz; 5World Health Organization, Country Office, Dar es Salaam P.O. Box 9292, Tanzania; mwengeew@who.int; 6United Nations Children’s Fund (UNICEF), Country Office, Dar es Salaam P.O. Box 4076, Tanzania; amphuru@unicef.org; 7Immunization Center, John Snow Inc. (JSI), 2733 Crystal Dr 4th Floor, Arlington, VA 22202, USA; nassor_mohamed@jsi.com; 8Management and Development for Health (MDH), Dar es Salaam P.O Box 79810, Tanzania; dkayabu@mdh.or.tz; 9Department of Epidemiology and Biostatistics, School of Public Health, Catholic University of Health and Allied Sciences, Mwanza P.O. Box 1464, Tanzania; ekonje@bugando.ac.tz

**Keywords:** CRS, active Rubella infection, Tanzania, vaccination

## Abstract

Rubella virus (RV) infection in susceptible women during the first trimester of pregnancy is associated with congenital Rubella syndrome (CRS). In countries where a vaccination program is implemented, active case surveillance is emphasized. This report documents the magnitude of active cases before and after vaccine implementation in Tanzania. A total of 8750 children and adolescents with signs and symptoms of RV infection were tested for Rubella IgM antibodies between 2013 and 2019 using enzyme immunoassay followed by descriptive analysis. The median age of participants was 3.8 (IQR: 2–6.4) years. About half (4867; 55.6%) of the participants were aged 1–5 years. The prevalence of RV active cases was 534 (32.6%, 95% CI: 30.2–34.9) and 219 (3.2%, 95% CI: 2.7–3.6) before and after vaccine implementation, respectively. Before vaccination, the highest prevalence was recorded in Pemba (78.6%) and the lowest was reported in Geita (15.6%), whereas, after vaccination, the prevalence ranged between 0.5% in Iringa and 6.5% in Pemba. Overall, >50% of the regions had a >90% reduction in active cases. The significant reduction in active cases after vaccine implementation in Tanzania underscores the need to sustain high vaccination coverage to prevent active infections and eventually eliminate CRS, which is the main goal of Rubella vaccine implementation.

## 1. Introduction

Rubella virus (RV) infection or ‘German measles’ is a communicable disease which is transmitted through aerosol, direct contact, or vertically from the mother to the fetus [1]. Rubella is an enveloped, single-stranded RNA positive-sense virus of the genus *Rubivirus* in the family *Togaviridae*. RV infection may present as an acute, mild, or asymptomatic illness. Symptoms of RV infection include rashes, low-grade fever, arthralgia, and lymphadenopathy [2]. Usually, symptoms begin with low-grade fever, malaise, and a morbilliform rash appearing simultaneously [3]. Rashes rarely last for more than 3 days, giving rise to the name ‘3 day measles’, and they usually start on the face, before extending to the trunk and extremities.

The disease affects mostly children, young adults including those of childbearing age, and pregnant women [1,4,5,6]. In most cases, the RV infection in children is self-limiting and rarely causes complications. Nevertheless, children usually harbor and spread the infection to the susceptible population including pregnant women and those who take care of them. The majority of adolescents including those of childbearing age in endemic areas such as Tanzania show high IgG seropositivity due to childhood exposure [5].

RV primary infection in susceptible pregnant women during the first trimester of pregnancy is often associated with an increased risk of adverse pregnancy outcomes such as abortions, stillbirths, and congenital Rubella syndrome (CRS), which is characterized by a triad of congenital heart diseases, bilateral cataracts, and hearing loss, among other features [7,8,9]. 

Teratogenic effects and clinical manifestations of the RV have been found to decrease with increasing gestational age at the time of vertical transmission [10]. Fetal manifestations in most cases are rare when maternal infection occurs after 16 weeks of gestation. CRS occurs in about 90–100% of the infants whose mothers were infected with the virus in the first trimester [11]. The rate of vertical transmission was found to drop to 25% in the late second trimester. However, the rate might increase to 35% around 27–30 weeks and rises to approximately 100% at 36 weeks of gestation [12,13,14].

Despite the global decrease in CRS cases, RV infection remains a public health concern in many low- and middle-income countries (LMICs) whereby more than 100,000 children are born with CRS each year [8,15,16]. In Africa, Rubella IgG seropositivity among pregnant women was found to range from 52.9% to 97.9% [17,18,19,20,21,22], with 2.1–47.1% of women susceptible to primary RV infection that could lead to CRS [22,23]. Among infected pregnant women, approximately 50% present with exanthematous skin lesions [17,24] which are not pathognomonic to RV infection. The overlapping of RV symptoms with other endemic viral infections such as Cytomegalovirus, Zika virus, and Parvovirus B19 poses a great challenge in clinical diagnosis, particularly in LMICs such as Tanzania; hence, serological diagnosis remains as an important tool for the identification of positive cases [25,26]. Although it has been a long-standing public health problem, there are no antiviral drugs available for treating RV infection or preventing vertical transmission that could lead to CRS. It should be noted that the cost of managing a CRS case in the upper- and middle-income countries has been found to range from 4261 to 57,010 USD per year and is estimated to be high in resource-limited countries [27]. Moreover, the role of passive immunization with immunoglobulin after maternal exposure is not feasible. Since there are no specific treatments available, vaccinating children, women of childbearing age, and multigravida women during the postpartum period significantly reduces the cases of CRS. 

The Rubella vaccination strategy aims to eliminate RV infection and CRS by interrupting transmission of RV through introduction of Rubella-containing vaccines (RCVs) into the country childhood routine immunization programs, supplemented by vaccination of older age groups during catch-up campaigns [28]. RCVs can be used as monovalent formulations or in combination with other antigens such as varicella, measles, and mumps (MMRV, MR, and MMR, respectively). Vaccination with a single dose of RCV provides ≥95% lifelong immunity against RV infection [28].

The Government of Tanzania through the Ministry of Health conducted a countrywide Measles/Rubella (MR) catch-up campaign targeting children aged 9 months to 15 years in October 2014, followed by the introduction of the MR vaccine into routine immunization in April 2015 [29]. Before introduction of the vaccine, the magnitude of RV active cases and CRS was not clear in Tanzania. However, 2014 data from the national sentinel sites reported 20 CRS cases, while, in 2016, only one case was reported [30]. Moreover, two studies from Mwanza conducted between 2014 and 2016 reported eight confirmed CRS cases [7,9]. Despite the reported high prevalence of RV worldwide, there are limited data on the magnitude of active cases and CRS in many LMICs where the virus is endemic [2,20,31,32,33,34]. This lack of relevant data has resulted in ineffective strategies for CRS control in these countries. The World Health Organization (WHO) recommends surveillance of RV active cases and CRS in the countries where Rubella vaccination programs are implemented [35]. Nevertheless, there is scarcity of these data in many LMICs with limited information on the magnitude of active cases. This report in Tanzania documents, for the first time, the magnitude of RV active cases involving different regions of Tanzania before and after vaccine implementation.

## 2. Materials and Methods

The United Republic of Tanzania, according to the 2012 census, documented that the population increased to 44.9 million with a projected population of 50.1 million, with 50.1% of the population aged between 0 and 17 years. Furthermore, the population aged between 0 and 17 years was found to range from 37.8% in Dar es Salaam to 57.8% in Simiyu (http://tanzania.countrystat.org/fileadmin/user_upload/countrystat_fenix/congo/docs/Population%20Distribution%20by%20Age%20and%20Sex%20Report-2012PHC.pdf (accessed on 27 June 2022)).

Between 2013 and 2019, a total of 9180 venous blood samples were collected from children and adolescents aged between 0 and 18 years with signs and symptoms of RV infection (fever and rash or history of fever and rash within 30 days of onset) [35]. Samples were collected as part of the surveillance of active RV cases in different regions of the United Republic of Tanzania. Blood samples were collected in a plain vacutainer tube (Becton Dickinson (E.A) Ltd. Health Care Product Dealers in Nairobi, Kenya) and transported to the respective laboratories of each region. In the laboratory, samples were centrifuged to obtain sera which were transported to the Immunization and Vaccine Development Program (VPD) surveillance unit for sorting and packaging, before being transported to the National Public Health Laboratory, Dar es Salaam for testing [36]. Sera were stored in freezers at −80 °C until processing; before processing, they were left to attain room temperature. Sera were processed as per standard operating procedures (SOP′s) and manufacturer′s instructions (Euroimmun IgM ELISA, Seekamp 31 23,560, Lubeck, Germany). Rubella-specific IgM antibodies were detected by indirect enzyme-linked immunosorbent assay (ELISA) as per the manufacturer′s instructions (Euroimmun IgM ELISA, Seekamp 31 23,560, Lubeck, Germany). The manufacturer determined the cutoff value by analyzing Rubella IgM antibodies in a panel of 500 healthy blood donors, concluding that a cutoff ratio of 1.0 can be used to classify 98.4% of cases as negative. The ratio values for the positive and negative controls and for each specimen were determined by dividing the specimen OD value by the cutoff calibrator OD. An index value of ≥1.1 was considered as Rubella IgM-seropositive, signifying active or recent infection. The sensitivity and specificity of the IgM ELISA used were 100%. 

A total of 430 samples were excluded from the final analysis because they had incomplete metadata or inconclusive results. Data of 8750 samples were entered into a computer using Microsoft Office Excel 2007 and analyzed using the STATA version 13 (College Station, TX, USA). Categorical variables were presented as proportions, and Pearson’s chi-square test was performed to observe the statistical differences among the various groups. Continuous variables were summarized as medians with interquartile ranges (IQRs). A *p*-value < 0.05 at a 95% confidence interval (CI) was considered statistically significant.

## 3. Results

The median age of the participants was 3.8 (IQR: 2–6.4) years. More than half (4867; 55.6%) of the participants were aged between 1 and 5 years, about one-third (2954; 33.7%) were aged between 2 and 18 years, and a total of 929 (10.6%) were infants. More than half (4598; 52.5%) of the participants were female. 

Overall, the seroprevalence of Rubella IgM antibodies before RV vaccine implementation (2013–October 2014) was nearly one-third of the total samples tested (534; 32.6%, 95% CI: 30.2–34.9%), whereas, after vaccine implementation, this value was 219 (3.2%, 95% CI: 2.7–3.6%) (Figure 1), translating to a 90.2% reduction in active cases of RV after vaccine implementation. 

When categorized by age groups, before vaccination the seroprevalence of RV active cases was significantly higher among 5–18 years age group than other age groups (*p* ≤ 0.001) while after vaccination it was significantly higher among infants than other age groups (*p* ≤ 0.001) (Table 1).

Further analysis revealed that more than 50% of the regions have >90% reduction of RV active cases (Table 2). Before vaccination the highest seroprevalence was observed in Pemba (78.6%) and the lowest in Geita (15.6%) while after vaccination the seroprevalence was found to range from 0.5% in Iringa to 6.5% in Pemba (Table 2, Figure 2).

## 4. Discussion

This is the first report involving 31 regions of the United Republic of Tanzania documenting the magnitude of active RV infection before and after Rubella vaccine implementation. The most important observation in this report is the significant reduction in the number of RV active cases after implementation of the Rubella vaccine in Tanzania. This observation is consistent with previous reports from high-income countries (HICs) where Rubella vaccine implementation reduced RV active cases. For example, in Germany, by the year 2018, the active RV incidence per 1,000,000 population was 0.7 cases [37], while, in the United Kingdom, between 2013 and 2021, IgM seropositivity rates among suspects using oral fluid test ranged from 0% to 3.2% [38]. The possible explanation for the observed low prevalence of RV active cases after vaccine implementation in Tanzania could be explained by high vaccination coverage in these regions, underscoring the need to maintain this coverage across the country to reach a goal of CRS elimination. Since its introduction into routine immunization in 2015 to 2021, the coverage of the first dose of the Measles/Rubella vaccine (MR1) at a national level has been ≥95% except for 2016 and 2021, where the coverage was 90% and 92%, respectively [39]. A single dose of RCV can provides ≥95% lifelong immunity to Rubella infection [28]. However, provision of a two-dose schedule for Rubella vaccine into routine immunization for infants is due to programmatic objectives since Rubella vaccines are often given in combination with other vaccines. Provision of a second dose of the Rubella vaccine has not been associated with change in the safety and efficacy profiles of the vaccine.

In this report, before vaccination, it was observed that IgM seroprevalence increased with age, in contrast to what was observed after vaccine implementation. The high seroprevalence in infants after vaccination could be explained by waning of maternal protection before vaccination at the age of 9 months as previously reported [9]. The findings from this report might be useful in emphasizing the importance of maintaining a high vaccine coverage across the country, especially in areas such as Pemba where the prevalence of active cases is above 5% after vaccine implementation (Figure 2). These findings are in agreement with the WHO regarding the surveillance of active RV cases after vaccine implementation, underscoring a need for sustained RV surveillance in other LMICs following vaccine implementation, which is important for early notifications of possible outbreaks.

Despite the observed impact of RV vaccine implementation in Tanzania among children below 5 years of age, there is a paramount need to embark on other strategies including screening and vaccinating women of childbearing age and susceptible pregnant women in the postpartum period. This has been found to be more cost-effective in developed countries than vaccinating all children and adolescents blindly. Moreover, this should be included in the antenatal package so that all pregnant women can be screened and those found to be susceptible can be identified and vaccinated in the postpartum period. This has been reported as an effective way of controlling CRS in developed countries [40]. Previous studies in Tanzania showed that up to 10% of women of childbearing age and pregnant women are susceptible to primary RV infection that might lead to CRS [5,20]. A combination of these strategies has been found to be effective in other countries that are in the CRS elimination phase.

Furthermore, to maintain the high two-dose MR vaccination coverage, the Tanzanian government has been implementing the following strategies: (i) provision of a first dose of RCV at or shortly after 9 months of age and providing a second dose at or shortly after 18 months of age, (ii) conducting a Reaching Every Child (REC) approach during routine immunization planning in the health facilities to trace and revaccinate all eligible children, (iii) periodic intensification of routine immunization to provide catch-up vaccinations for all children that were not reached at their appropriate age for vaccination, (iv) intensification of immunization activities during African immunization week to reach children who were not reached during routine vaccination, (v) integration of vaccination activities with other interventions such as nutrition screening, vitamin A, and deworming campaigns, (vi) promotion of community involvement through sensitization meetings to maintain vaccine confidence in the community, and (vii) provision of an opportunity for measles vaccination through Supplementary Immunization Activities (SIAs) at intervals of 2–4 years targeting children up to 10 years depending on the epidemiological situation [4] and the population immunity, as measured by the vaccination coverage levels. The last SIA was conducted in October 2019.

## 5. Conclusions

Routine rubella vaccination in Tanzania has resulted in overall 90.2% reduction in reported RV active cases, which may have led to a significant reduction in CRS. The Tanzanian government should sustain routine RV vaccination, as well as high vaccination coverage, to reach the CRS elimination phase. In-depth characterization of seasonal patterns linked with sero-epidemiology should also be established by further studies.

## Figures and Tables

**Figure 1 vaccines-10-01188-f001:**
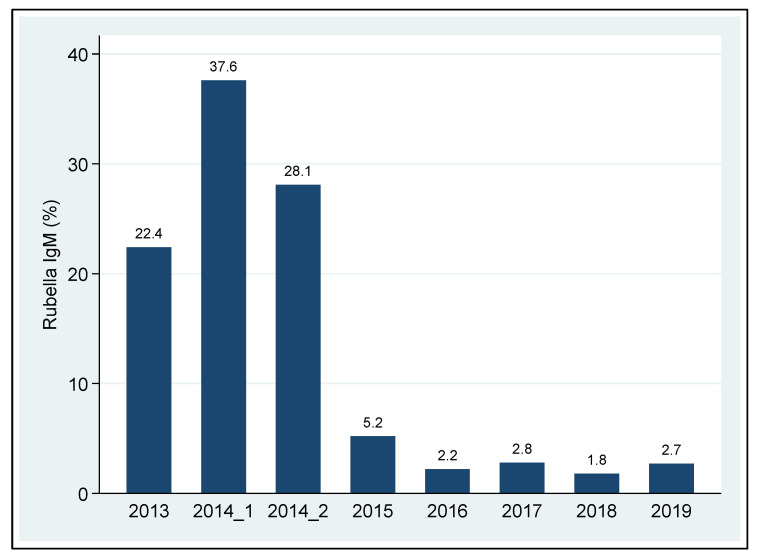
Prevalence of active Rubella cases by years in Tanzania.

**Figure 2 vaccines-10-01188-f002:**
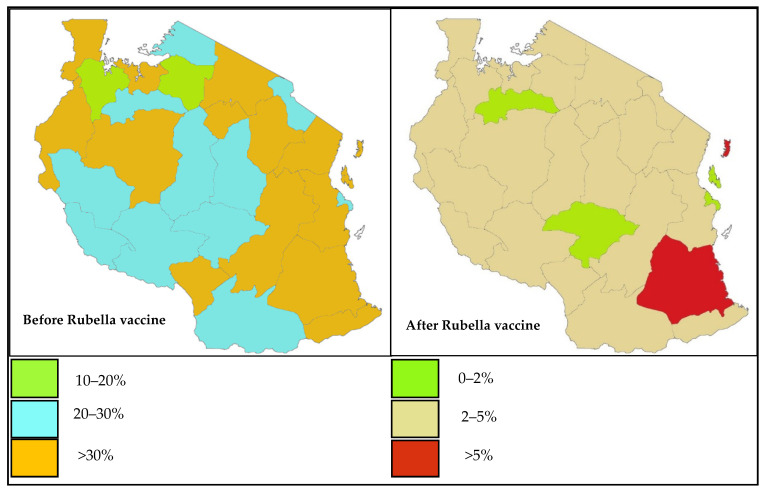
Map of the United Republic of Tanzania showing magnitude of Rubella virus active cases percentage IgM positivity among samples tested) before and after vaccine implementation (credit: CUHAS).

**Table 1 vaccines-10-01188-t001:** Active Rubella cases by age groups before and after vaccine implementation in Tanzania.

SN	Age Category (years)	Before	After
1	0–1	71/101 (16.8%)	48/828 (5.8%)
2	1–5	169/645 (26.2%)	103/4222 (2.4%)
3	5–18	348/892 (39.1%)	68/2062 (3.3%)
	*p*-value	*p* < 0.01	*p* < 0.01

**Table 2 vaccines-10-01188-t002:** Rubella virus active cases before and after vaccine implementation by regions.

SN	Region	Sample Tested	Cases before Vaccination (%)	Sample Tested	Cases after Vaccination (%)	Reduction (%)
1	Arusha	80	33 (41.3)	208	6 (2.9)	93.0
2	Dar es salaam	111	32 (29.8)	375	6 (1.6)	94.6
3	Dodoma	102	21 (20.6)	321	8 (2.5)	87.9
4	Geita	19	21 (15.8)	266	8 (3)	81.0
5	Iringa	40	9 (22.5)	200	1 (0.5)	97.8
6	Kagera	68	26 (38.2)	351	10 (2.8)	92.7
7	Katavi	25	6 (24)	207	10 (4.8)	80.0
8	Kigoma	77	30 (39)	362	9 (2.5)	93.6
9	Kilimanjaro	79	23 (29.1)	188	7 (3.7)	87.3
10	Lindi	54	17 (31.5)	160	9 (5.6)	82.2
11	Manyara	41	19 (46.3)	194	7 (3.6)	92.2
12	Mara	55	16 (29.1)	262	6 (2.3)	92.1
13	Mbeya	120	22 (18.3)	361	10 (2.8)	84.7
14	Morogoro	35	13 (37.1)	241	13 (5.4)	85.4
15	Mtwara	81	26 (32.1)	604	23 (3.8)	88.2
16	Mwanza	134	50 (37.3)	284	9 (3.2)	91.4
17	Njombe	39	15 (38.5)	203	6 (2.9)	92.5
18	* Pemba	14	11 (78.6)	31	2 (6.4)	91.9
19	Pwani	90	36 (40)	210	5 (2.4)	94.0
20	Rukwa	24	6 (25)	147	7 (4.7)	81.2
21	Ruvuma	39	10 (25.6)	196	6 (3)	88.3
22	Shinyanga	28	7 (25)	258	3 (1.2)	95.2
23	Simiyu	34	6 (17.6)	276	13 (4.7)	73.3
24	Singida	47	14 (29.8)	350	6 (1.7)	94.3
25	Songwe	10	2 (20)	196	6 (3.1)	84.5
26	Tabora	49	20 (40.8)	224	8 (3.6)	91.2
27	Tanga	135	58 (42.9)	337	14 (4.1)	90.4
28	** Unguja	8	3 (37.5)	97	1 (1)	97.3
Total		1638	534 (32.6)	7112	219 (3.1)	90.5

* Includes South Pemba and North Pemba. ** Includes Zanzibar Central/South, Zanzibar North, and Zanzibar Urban/West.

## Data Availability

All data are included in the manuscript.

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
