# Peer review of "Reduction in Rubella Virus Active Cases among Children and Adolescents after Rubella Vaccine Implementation in Tanzania: A Call for Sustained High Vaccination Coverage"

_vaccines, 2022, doi:10.3390/vaccines10081188_

Round 1
Reviewer 1 Report
The authors generally report the RV IgM+ percentage in the children and adolescent populations before and after vaccine implementation to emphasize the importance the vaccination to RV infection. Here are my questions.
1. When specifically did the vaccine implementation start? Did it start in the whole country at the same time? In figure 1, it seems the number dropped significantly between 2014_2 and 2015.
2. For Figure 2, does the color code represent the average or median numbers percentage of IgM+ samples in different regions among the years before and after vaccine implementation? The authors should specify what numbers they used for figure 2 in the legend.
3. In Figure 2, although all the region have dramatic decrease in the percentage, some regions are still red, as the authors mentioned in the discussion. It would be more helpful to the readers, if the authors can explain other potential actions that the government and public can do to prevent the infection, besides the vaccination.
4. Were these samples also tested for viral titers? Or just the IgM. The correlation between the viral dose and the antibody level would be more informative.
5. Moreover, it would be great if the authors can clarify if there is any improvement the government can do to promote the vaccination in the country and what stage of age would be the best for vaccination.
6. Lastly, there are some typos in the manuscript.
Author Response
EIC,
The file has been uploaded

Reviewer 2 Report
The manuscript describes the magnitude of active cases of RV before and after vaccine implementation in a total of 8750 samples, including children and adolescents, with signs and symptoms of RV infection from different regions of Tanzania. Considering no specific treatment is available, the only strategy is immunization programs supplemented by vaccination. This investigation is important as a significantly reduced number of RV infections was observed after vaccination implementation. The manuscript needs an upgrade before being accepted for publication.
Major points:
Introduction: It could be interesting to describe demographic data from the countries studied.
Material and Methods: It is not clear how many samples were collected from 31 regions in Tanzania. Please be clear and introduce the information (These data are possible to see only in the results). It should be nice put, also how the cut-off was calculated.
Results: The interpretations and results are consistent. The tables and figures are adequate, and the data is informative. However, the graphic presentation could be improved. Please introduce the graphic (using a graph prism, for example) of Figure 1.
Discussion: Concerning the discussion, it clearly and accurately describes the article's content. The references are adequate, and the results justify the conclusions.
Author Response
EIC,
The file has been uploaded

Round 2
Reviewer 2 Report
The authors accepted all the suggestions, and the manuscript suffered a good impruvment. We now suggest your acceptance.